# STOCHASTIC UNROLLED FEDERATED LEARNING

## ABSTRACT

Algorithm unrolling has emerged as a learning-based optimization paradigm that unfolds truncated iterative algorithms in trainable neural-network optimizers. We introduce Stochastic UnRolled Federated learning (SURF), a method that expands algorithm unrolling to a federated learning scenario. Our proposed method tackles two challenges of this expansion, namely the need to feed whole datasets to the unrolled optimizers to find a descent direction and the decentralized nature of federated learning. We circumvent the former challenge by feeding stochastic mini-batches to each unrolled layer and imposing descent constraints to mitigate the randomness induced by using mini-batches. We address the latter challenge by unfolding the distributed gradient descent (DGD) algorithm in a graph neural network (GNN)-based unrolled architecture, which preserves the decentralized nature of training in federated learning. We theoretically prove that our proposed unrolled optimizer converges to a near-optimal region infinitely often. Through extensive numerical experiments, we also demonstrate the effectiveness of the proposed framework in collaborative training of image classifiers.

## 1 INTRODUCTION

Federated learning is a distributed learning paradigm in which a set of low-end devices aim to collaboratively train a global statistical model. A growing body of work, e.g., (Lian et al., 2015; McMahan et al., 2016; Li et al., 2020b), has deployed a server in the network to facilitate reaching consensus among the agents, which creates a communication bottleneck at the server and requires high bandwidth when the number of agents grows large. To alleviate these challenges, another line of work that traces back to decentralized optimization (Nedic and Ozdaglar, 2009; Wei and Ozdaglar, 2012; Wu et al., 2017) has instead investigated peer-to-peer communication, eliminating the role of central servers in the network. These server-less federated learning frameworks compromise communication efficiency and convergence rates (Vanhaesebrouck et al., 2017; Liu et al., 2022a;b). The slow convergence of these methods arises as a practical challenge since it greatly outweighs the capacity of resource- and energy-constrained devices.

Algorithm unrolling has recently emerged as a learning-to-optimize paradigm that unfolds iterative algorithms via learnable neural networks, achieving state-of-the-art performance in many applications such as computer vision (Zhang and Ghanem, 2018), policy learning (Marino et al., 2021), and computational biology (Cao et al., 2019) to name a few. The key reported advantage of learning the parameters of standard algorithms is achieving much faster convergence without sacrificing performance (Monga et al., 2021). The fast-convergence advantage of unrolling could potentially surmount the challenges faced by low-end devices when collaboratively training deep models.

Nevertheless, the convergence analysis of unrolled algorithms is still in its infancy. One approach, known as safeguarding, has been proposed in (Heaton et al., 2023; Shen et al., 2021; Moeller et al., 2019; Liu et al., 2021b), where the estimate made by a certain layer is considered only if it is in a descent direction; otherwise, it is replaced with an estimate of the classic iterative algorithm. Some other studies, e.g., (Xie et al., 2019; Chen et al., 2018b), provided theoretical proofs for the existence of unrolled networks that converge to the optimal; however, they do not provide methods for finding these convergent networks. To resolve this issue, (Liu and Chen, 2019; Abadi et al., 2016) learn fewer parameters of the standard algorithm, which limits the network's expressivity. These theoretical proofs are also tailored for specific optimization algorithms and do not generalize to other algorithms. That leaves many existing and potential unrolled networks without convergence guarantees. The lack of convergence guarantees precludes perceiving unrolled networks as descent

algorithms, which, in turn, raises concerns about their generalizability and usability in safety-critical applications.

In this paper, we introduce unrolling to server-less federated learning and tackle the issue of lacking convergence guarantees. We pose unrolled architectures for decentralized federated learning as stochastic descent algorithms under a proposed training framework called Stochastic UnRolled Federated learning (SURF). To achieve this, SURF imposes *descending constraints* on the training procedure of the unrolled architectures. We theoretically prove that the unrolled architecture trained using SURF is a stochastic descent optimizer that converges to a near-optimal region of the loss function of the federated problem.

In addition, SURF is agnostic to the underlying algorithm being unrolled. As a demonstration of its effectiveness, we unroll one of the earliest decentralized optimization algorithms in the literature known as decentralized gradient descent (DGD) (Nedic and Ozdaglar, 2009). The unrolled architecture is parameterized with the help of graph neural networks (GNNs), hence preserving the decentralized nature of the problem.

In summary, our contributions are as follows:

- We develop stochastic unrolling for decentralized federated learning, which encourages faster convergence compared to other standard algorithms (as depicted in Figure 1).

- We force the unrolled architectures to converge by imposing descending constraints within our training framework, SURF. We theoretically (see Theorem 2) and empirically (see Figure 2) prove that an unrolled optimizer trained via SURF converges to a near-optimal region and is, therefore, guaranteed to generalize to in-distribution datasets.

- We empirically show (see Figure 3) that the imposed constraints provide the unrolled optimizers with robustness to perturbations caused by a lack of synchrony between the agents.

One of the advantages that SURF provides is shifting where a neural network is trained using gradient descent, i.e., moving from training a neural network online in a decentralized manner to training an unrolled network offline. This moves the demanding hardware training requirements from low-end devices to more powerful offline servers. The downside of using unrolling in training neural networks is that the size of the unrolled network is typically much larger than the original one. Therefore, we envision SURF as a method that complements other federated learning frameworks without necessarily replacing them. Particularly, SURF best suits problems of training relatively lightweight models on resource- and energy-limited devices, where fast convergence is a priority.

## 2 RELATED WORK

**Server-less Federated Learning.** There have been many efforts in recent years to enable federated learning without the aid of a server, e.g., (Kalra et al., 2023; Sun et al., 2023; Wang et al., 2022; Tedeschini et al., 2022; Ye et al., 2022; Wink and Nochta, 2021) to name a few. These efforts have benefited from the advances in decentralized algorithms, such as decentralized SGD (Koloskova et al., 2020; Wang and Joshi, 2021), asynchronous decentralized SGD (Lian et al., 2018), and alternating direction method of multipliers (ADMM) (Wei and Ozdaglar, 2012; Shi et al., 2014). Our proposed method deviates from these studies in that we use a meta approach to learn the optimizer instead of using state-of-the art optimizers.

**Learning to Optimize/Learn (L2O/L2L).** Our work is mostly related to the broad research area of L2O (Chen et al., 2021b), which aims to automate the design of optimization methods by training optimizers on a set of training problems. L2O has achieved notable success in many optimization problems including $\ell_1$-regularization (Gregor and LeCun, 2010), neural-network training (Andrychowicz et al., 2016; Ravi and Larochelle, 2016), minimax optimization (Shen et al., 2021), and black-box optimization (Chen et al., 2017) among many others.

Prior work in L2O can be divided into two categories; model-free and model-based optimizers. Model-free L2O aims to train an iterative update rule that does not take any analytical form and relies mainly on general-purpose recurrent neural network (RNNs) and long short-term memory networks (LSTMs) (Andrychowicz et al., 2016; Chen et al., 2017; Lyu et al., 2017; Wichrowska et al., 2017; Xiong and Hsieh, 2020; Jiang et al., 2021). Model-based L2O, on the other hand, provides compact,

interpretable learning networks by taking advantage of both model-based algorithms and data-driven learning paradigms (Gregor and LeCun, 2010; Greenfeld et al., 2019). As part of this category, algorithm unrolling aims to unroll the hyperparameters of a standard iterative algorithm in a neural network to learn them. The seminal work (Gregor and LeCun, 2010) unrolled iterative shrinkage thresholding algorithm (ISTA) for sparse coding problems. Following (Gregor and LeCun, 2010), many other algorithms have been unrolled, including, but not limited to, projected gradient descent (Giryes et al., 2018), the primal-dual hybrid gradient algorithm (Jiu and Pustelnik, 2020; Cheng et al., 2019), and Frank-Wolfe (Liu et al., 2019).

Learning to learn (L2L) refers to frameworks that extend L2O to training neural networks in small data regimes, e.g., few-shot learning (Triantafillou et al., 2020). Learning to learn has strong ties to meta-learning, but they differ in their ultimate goal; meta-learning, e.g., model-agnostic meta-learning (MAML) (Finn et al., 2017), aims to learn an initial model that can be fine-tuned in a few gradient updates, whereas L2L aims to learn the gradient update and the step size. General purpose LSTM-based models, e.g., (Ravi and Larochelle, 2016; Andrychowicz et al., 2016; Li et al., 2017) are the most popular among L2L models.

**Algorithm Unrolling in Distributed Problems.** Algorithm unrolling has also been introduced to distributed optimization problems with the help of graph neural networks (GNNs). One of the first distributed algorithms to be unrolled was weighted minimum mean-square error (WMMSE) (Shi et al., 2011), which benefited many applications including wireless resource allocation (Chowdhury et al., 2021; Li et al., 2022) and multi-user multiple-input multiple-output (MU-MIMO) communications (Hu et al., 2021; Zhou et al., 2022; Ma et al., 2022; Pellaco and Jaldén, 2022; Schynol and Pesavento, 2022; 2023). Several other distributed unrolled networks have been developed for graph signal denoising (Chen et al., 2021a; Nagahama et al., 2021), graph topology inference (Pu et al., 2021) and computer vision (Lin et al., 2022), among many others. In our work, we follow the lead of these studies and rely on GNNs to unroll DGD for federated learning. To the best of our knowledge, our work is the first to use algorithm unrolling in a federated learning setting.

## 3 PROBLEM FORMULATION

Consider a network of $n$ agents that periodically coordinate to train a statistical model $\Psi : \mathcal{X} \to \mathcal{Y}$, parameterized by $\mathbf{w} \in \mathbb{R}^d$, to fit a pair of random variables $\mathbf{x} \in \mathcal{X}$ and $\mathbf{y} \in \mathcal{Y}$ jointly distributed according to data distribution $\mathcal{D}$. To achieve this goal, the agents communicate over a server-less network, represented by an undirected connected graph $\mathcal{G} = (\mathcal{V}, \mathcal{E})$, where $\mathcal{V} = \{1, \dots, n\}$ denotes the set of nodes and $\mathcal{E} \subseteq \mathcal{V} \times \mathcal{V}$ denotes the set of edges. The graph is associated with a real symmetric matrix $\mathbf{S} \in \mathbb{R}^{n \times n}$, which has a non-zero entry iff either $(i, j) \in \mathcal{E}$ or $i = j$. We denote the neighborhood of node $i$ by $\mathcal{N}_i = \{j \in \mathcal{V} | (i, j) \in \mathcal{E}\} \cup \{i\}$, within which the agent transmits its current estimate of $\mathbf{w}$. Under these assumptions and notation, we next formally define the federated learning problem and the algorithm-unrolling approach we adopt to solve it.

### 3.1 FEDERATED LEARNING

The decentralized federated learning problem can be cast as the separable, constrained problem

$$
\begin{aligned}
\min_{\mathbf{w}_1, \dots, \mathbf{w}_n \in \mathbb{R}^d} \quad & f(\mathbf{W}) := \frac{1}{n} \sum_{i=1}^{n} \mathbb{E}[\ell(\Psi(\mathbf{x}_i; \mathbf{w}_i), \mathbf{y}_i)], \\
\text{s.t.} \quad & \mathbf{w}_i = \frac{1}{|\mathcal{N}_i|} \sum_{j \in \mathcal{N}_i} \mathbf{w}_j, \quad \forall i \in \mathcal{V},
\end{aligned}
\tag{FL}
$$

where $\mathbf{w}_i$ is a *local* version of the global variable $\mathbf{w}$ stored at agent $i$, and all $\mathbf{w}_i$'s are arranged in the rows of the matrix $\mathbf{W} \in \mathbb{R}^{n \times d}$. The (FL) problem aims to minimize a global objective function $f : \mathbb{R}^{n \times d} \to \mathbb{R}$ that is the average of some local loss functions $\ell : \mathcal{X} \times \mathcal{Y} \to \mathbb{R}$. The local objective is to train a statistical model $\Psi$ in a supervised mode, in which each agent $i \in \mathcal{V}$ has access to local data distributed according to an unknown probability distribution $\mathcal{D}$ over the space of data pairs $\mathbf{x}_i \in \mathcal{X}$ and $\mathbf{y}_i \in \mathcal{Y}$. Moreover, the (FL) problem deploys constraints that require each local variable to stay equal to the average of the direct neighbors' local variables. When satisfied, the average constraints boil down to constraints of the form $\mathbf{w}_i = \mathbf{w}_j$ for all $i$ and $j$ due to the connectivity and symmetry of the graph, hence leading to consensus among agents.

## 3.2 ALGORITHM UNROLLING

To solve (FL), we opt for an L2O approach, where we unfold an arbitrary standard decentralized optimizer in the layers of an unrolled model $\boldsymbol{\Phi} \in \mathcal{H}$, which we refer to as the *optimizer*. The optimizer $\boldsymbol{\Phi}$ is parameterized by a sequence of learnable parameters $\boldsymbol{\theta} = \{\boldsymbol{\theta}_l\}_{l=1}^L$, each of which mimics the parameters of one iteration in the standard algorithm. Learning these parameters can then be cast as the bi-level optimization problem

$$\underset{\boldsymbol{\theta}}{\operatorname{argmin}} \quad \mathbb{E}\big[f(\boldsymbol{\Phi}(\boldsymbol{\vartheta}; \boldsymbol{\theta}))\big]$$
$$\text{s.t.} \quad \mathbf{W}_l = \phi(\mathbf{W}_{l-1}, \boldsymbol{\vartheta}; \boldsymbol{\theta}_l), \quad l = 1, \ldots, L, \qquad \text{(Optimizer)}$$
$$\mathbf{W}_L = \boldsymbol{\Phi}(\boldsymbol{\vartheta}; \boldsymbol{\theta}),$$

where $\mathbf{W}_l$ is the output of the $l$-th layer, $L$ is the number of layers, and $\boldsymbol{\vartheta}$ is a dataset of data pairs $(\mathbf{x}, \mathbf{y}) \sim \mathcal{D}$ divided among $n$ agents. The initial estimate $\mathbf{W}_0$ is drawn from a Gaussian distribution $\mathcal{N}(\boldsymbol{\mu_0}, \sigma_0^2 \mathbf{I})$. The lower-level problem iterates the update rule $\phi$ of the standard decentralized algorithm to find the optimal of (FL). On the other hand, the upper-level problem finds the optimal parameters of $\phi$ at each layer $l$ that ensure the network's output $\mathbf{W}_L$ is a stationary point of $f$.

The statistical nature of the objective function of the upper-level problem indicates that it is a *meta-training* problem, where the learnable parameters are trained to fit a distribution of similar *tasks*. For example, the unrolled network $\boldsymbol{\Phi}$ can be trained over a meta-training dataset that contains image datasets with different label distributions. When the trained unrolled optimizer is executed on a query dataset sampled from the same task distribution, each layer is supposed to take a descent direction over the manifold constructed by this query dataset. Two challenges, however, become evident: i) lack of convergence guarantees of the unrolled optimizer hinders its generalizability to query datasets, and ii) a whole dataset needs to be fed to each layer of the unrolled network in order to define the manifold we optimize over. In the following section, we tackle these two issues in our proposed training method, SURF.

## 4 PROPOSED METHOD

To tackle the aforementioned challenges, we introduce Stochastic UnRolled Federated learning, or SURF, a training method that provides unrolled decentralized architectures with convergence guarantees. SURF guarantees convergence by imposing supermartingale-like *descending constraints* at each unrolling layer. Moreover, it resolves the latter challenge of massive and variable-size query datasets using *stochastic* unrolling, where we feed each layer $l \in \{1, \ldots, L\}$ of the unrolled network with a *fixed-size batch* $\mathbf{B}_l$ sampled independently and uniformly at random from the dataset $\boldsymbol{\vartheta}$.

The stochastic unrolled federated learning problem can then be formulated as

$$\min_{\boldsymbol{\theta}} \quad \mathbb{E}\big[f(\boldsymbol{\Phi}(\boldsymbol{\vartheta}; \boldsymbol{\theta}))\big]$$
$$\text{s.t.} \quad \mathbb{E}\big[\|\nabla f(\mathbf{W}_l)\| - (1 - \epsilon)\|\nabla f(\mathbf{W}_{l-1})\|\big] \leq 0, \ \forall l = 1, \ldots, L, \qquad \text{(SURF)}$$
$$\mathbf{W}_l = \phi(\mathbf{W}_{l-1}, \mathbf{B}_l; \boldsymbol{\theta}_l), \quad \forall l = 1, \ldots, L,$$

where $\nabla$ denotes stochastic gradients, $\| \cdot \|$ is the Frobenius norm, and $\epsilon \in (0, 1)$. The expectation in the constraints accounts for the randomness in the initial estimate $\mathbf{W}_0$, the mini-batch $\mathbf{B}_l$, and the task distribution. The descending constraints, therefore, force the gradients to decrease over the layers despite the randomness introduced by relying on a few data points to estimate a descent direction. Intuitively, this would stimulate the unrolled optimizer to converge to a *local* optimal, i.e., $\mathbf{W}_l \to \mathbf{W}^*$, on average. Observe that the loss function $f$ is probably non-convex with respect to $\mathbf{w}_i$ (see (FL)), and therefore, we consider convergence to local minima.

It is worth noting that the stochastic unrolling we propose in this paper is crucially different from the stochastic approximations used in (Ravi and Larochelle, 2016; Andrychowicz et al., 2016; Liu et al., 2021a). In these studies, the gradients of the objective function $f$ are approximated over a mini-batch before they are fed to the unrolled layers, thereby reducing the computational complexity of these algorithms. However, these studies have overlooked the effects of the stochastic (noisy) approximations of the gradients on the performance of their algorithms. In (SURF), we feed the mini-batches directly to the unrolled network in order to learn a descent direction instead of computing it and tackle the uncertainty in these estimated directions using the descending constraints.

## 4.1 PROBABLY, APPROXIMATELY CORRECT SOLUTION TO (SURF)

To find the minimizer of (SURF), we leverage the constrained learning theory (CLT) (Chamon et al., 2022) by appealing to its dual problem. We formulate the latter by finding the saddle point of the Lagrangian function

$$\mathcal{L}(\boldsymbol{\theta}, \boldsymbol{\lambda}) = \mathbb{E}\big[f(\boldsymbol{\Phi}(\boldsymbol{\vartheta}; \boldsymbol{\theta}))\big] + \sum_{l=1}^{L} \lambda_l \mathbb{E}\big[\|\nabla f(\mathbf{W}_l)\| - (1 - \epsilon) \|\nabla f(\mathbf{W}_{l-1})\|\big], \tag{1}$$

where $\boldsymbol{\lambda} \in \mathbb{R}_+^L$ is a vector collecting the dual variables $\lambda_l$. Since a closed-form of the expectation over an unknown distribution is unattainable, we resort to the *empirical* Lagrangian function

$$\widehat{\mathcal{L}}(\boldsymbol{\theta}, \boldsymbol{\lambda}) = \widehat{\mathbb{E}}\big[f(\boldsymbol{\Phi}(\boldsymbol{\vartheta}; \boldsymbol{\theta}))\big] + \sum_{l=1}^{L} \lambda_l \widehat{\mathbb{E}}\big[\|\nabla f(\mathbf{W}_l)\| - (1 - \epsilon) \|\nabla f(\mathbf{W}_{l-1})\|\big], \tag{2}$$

where $\widehat{\mathbb{E}}$ denotes the sample mean evaluated over $Q$ realizations. The empirical dual problem can then be cast as

$$\widehat{D}^* = \max_{\boldsymbol{\lambda} \in \mathbb{R}_+^L} \min_{\boldsymbol{\theta}} \widehat{\mathcal{L}}(\boldsymbol{\theta}, \boldsymbol{\lambda}). \tag{3}$$

Equation (3) is an unconstrained optimization problem that can be solved by alternating between minimizing the Lagrangian with respect to $\boldsymbol{\theta}$ for a fixed $\boldsymbol{\lambda}$ and then maximizing over the latter, as described in Algorithm 1.

Nevertheless, (3) is not equivalent to (SURF) due to the non-convexity of the problem and the empirical gap induced by replacing the statistical expectations with empirical ones. A precise characterization of the gap between the two problems is provided by CLT under the following assumptions:

**Assumption 1.** The loss function $f(\cdot)$ in (SURF) and the gradient norm $\|\nabla f(\cdot)\|$ are both bounded and $M$-Lipschitz continuous functions.

**Assumption 2.** Let $\widehat{\mathbb{E}}$ be the sample mean evaluated over $Q$ realizations. Then there exists $\zeta(Q, \delta) \geq 0$ that is monotonically decreasing with $Q$, for which it holds with probability $1 - \delta$ that

1. $|\mathbb{E}[f(\boldsymbol{\Phi}(\boldsymbol{\vartheta}; \boldsymbol{\theta}))] - \widehat{\mathbb{E}}[f(\boldsymbol{\Phi}(\boldsymbol{\vartheta}; \boldsymbol{\theta}))]| \leq \zeta(Q, \delta)$, and

2. $|\mathbb{E}[\|\nabla f(\mathbf{W}_l(\boldsymbol{\vartheta}; \boldsymbol{\theta}))\|] - \widehat{\mathbb{E}}[\|\nabla f(\mathbf{W}_l(\boldsymbol{\vartheta}; \boldsymbol{\theta}))\|]| \leq \zeta(Q, \delta)$ for all $l$ and all $\boldsymbol{\theta} \in \mathbb{R}^p$.

**Assumption 3.** Let $\phi_l \circ \ldots \circ \phi_1 \in \mathcal{P}_l$ be a composition of $l$ unrolled layers parameterized by $\boldsymbol{\theta}_{1:l}$ and $\overline{\mathcal{P}}_l = \overline{conv}(\mathcal{P}_l)$ be the convex hull of $\mathcal{P}_l$. Then, for each $\overline{\phi}_l \circ \ldots \circ \overline{\phi}_1 \in \overline{\mathcal{P}}$ and $\nu > 0$, there exists $\boldsymbol{\theta}_{1:l}$ such that $\mathbb{E}\big[|\phi_l \circ \ldots \circ \phi_1(\mathbf{W}_0, \boldsymbol{\vartheta}; \boldsymbol{\theta}_{1:l}) - \overline{\phi}_l \circ \ldots \circ \overline{\phi}_1(\mathbf{W}_0, \boldsymbol{\vartheta})|\big] \leq \nu$ for all $l$.

**Assumption 4.** There exists $\boldsymbol{\Phi} \in \mathcal{H}$ that is strictly feasible, i.e., $\mathbb{E}\big[\|\nabla f(\mathbf{W}_l)\| - (1 - \epsilon) \|\nabla f(\mathbf{W}_{l-1})\|\big] < -M\nu, \forall l$, with $M$ and $\nu$ as in Assumptions 1 and 3.

**Assumption 5.** The conditional distribution $p(\boldsymbol{\vartheta}|\mathbf{W})$ is non-atomic for all $\mathbf{W}$.

CLT asserts that the gap between the two problems is affected by a smoothness constant $M$, the richness of the parameterization $\boldsymbol{\theta}$, and the sample complexity.

**Theorem 1** (CLT (Chamon et al., 2022)). *Let $P^*$ be the optimal value of* (SURF) *and $(\boldsymbol{\theta}^*, \boldsymbol{\lambda}^*)$ be a stationary point of* (3). *Under Assumptions 1- 5, it holds, for some constant $\rho$, that*

$$|P^* - \widehat{D}^*| \leq M\nu + \rho\, \zeta(Q, \delta), \text{ and} \tag{4}$$

$$\mathbb{E}\big[\|\nabla f(\mathbf{W}_l)\| - (1 - \epsilon) \|\nabla f(\mathbf{W}_{l-1})\|\big] \leq \zeta(Q, \delta), \quad \forall l, \tag{5}$$

*with probability $1 - \delta$ each and with $\rho \geq \max\{\|\boldsymbol{\lambda}^*\|, \|\overline{\boldsymbol{\lambda}}^*\|\}$, where $\overline{\boldsymbol{\lambda}}^* = \operatorname{argmax}_{\boldsymbol{\lambda}} \min_{\boldsymbol{\theta}} \mathcal{L}(\boldsymbol{\theta}, \boldsymbol{\lambda})$.*

The assumptions under which this result holds can be satisfied easily in practice. Assumption 1 requires the loss function and its gradient to be smooth and bounded. Assumption 2 identifies the sample complexity, which is a common assumption when handling statistical models. Moreover, Assumption 3 forces the parameterization $\boldsymbol{\theta}_l$ to be sufficiently rich at each layer $l$, which is guaranteed by modern deep learning models. Assumption 4 ensures that the problem is feasible and well

---

**noend 1** Primal-Dual Training Algorithm for (SURF)

---

1: **Input:** Meta-training dataset $\mathbf{D} = \{\vartheta_m\}_{m=1}^M$.
2: Initialize $\boldsymbol{\theta} = \{\boldsymbol{\theta}_l\}_{l=1}^L$ and $\boldsymbol{\lambda} = \{\boldsymbol{\lambda}_l\}_{l=1}^L$.
3: **for** each epoch **do**
4:     **for** each batch **do**
5:         Sample a dataset from $\mathbf{D}$ and compute $\widehat{\mathcal{L}}(\boldsymbol{\theta}, \boldsymbol{\lambda})$ as in (2).
6:         **for** $l = 1, \ldots, L$ **do**
7:             Update variables at layer $l$:

$$\boldsymbol{\theta}_l \leftarrow [\boldsymbol{\theta}_l - \mu_\theta \nabla_{\boldsymbol{\theta}_l} \widehat{\mathcal{L}}(\boldsymbol{\theta}, \boldsymbol{\lambda})], \tag{6}$$

$$\boldsymbol{\lambda}_l \leftarrow [\boldsymbol{\lambda}_l + \mu_\lambda \nabla_{\boldsymbol{\lambda}_l} \widehat{\mathcal{L}}(\boldsymbol{\theta}, \boldsymbol{\lambda})]_+. \tag{7}$$

8: **Return:** $\boldsymbol{\theta}_l^* \leftarrow \boldsymbol{\theta}_l, \forall l \in \{1, \ldots, L\}$.

---

posed, which is guaranteed since (SURF) mimics the parameters of a standard iterative solution. Finally, Assumption 5 can be satisfied using data augmentation.

Theorem 1 indicates that an unrolled optimizer trained via Algorithm 1 is a probably near-optimal, near-feasible solution to (SURF). Each unrolled layer takes a step in a descent direction with probability $1 - \delta$, where $\delta$ depends on the size of the training dataset. As a consequence, the trained unrolled optimizer can be considered as a stochastic descent algorithm.

### 4.2 Convergence Guarantees

Finding a probably approximately correct solution $\boldsymbol{\theta}^*$ does not directly guarantee its capability to generate a sequence of layers' outputs $\mathbf{W}_l$ that converges to the optimal solution of (FL). This is because this convergence requires (almost) all the descending constraints to be satisfied, which has a decreasing probability $(1 - \delta)^L$ with the number of layers $L$ despite the fact that these constraints are statistically independent. In Theorem 2, we prove that the trained unrolled optimizer indeed converges to a near-optimal region infinitely often if Assumption 1 holds.

**Theorem 2.** *Given are a probably approximately correct unrolled optimizer $\boldsymbol{\theta}^*$ that satisfies* (5) *with a probability $1 - \delta$ and generates a sequence of random variables $\mathbf{W}_1, \mathbf{W}_2, \ldots$ at the outputs of its layers. Then, under Assumption 1, it holds that*

$$\lim_{l \to \infty} \mathbb{E}\left[ \min_{k \le l} \|\nabla f(\mathbf{W}_k)\| \right] \le \frac{1}{\epsilon} \left( \zeta(Q, \delta) + \frac{\delta M}{1 - \delta} \right) \quad a.s. \tag{8}$$

*with $\zeta(Q, \delta)$ as described in Assumption 2.*

The proof constructs a stochastic process $\alpha_l$ that keeps track of the gradient norm until it drops below $\frac{1}{\epsilon}\left(\zeta(Q, \delta) + \frac{\delta M}{1 - \delta}\right)$ and shows that $\alpha_l$ converges almost-surely using the supermartingale convergence theorem (Robbins and Siegmund, 1971). The detailed proof of Theorem 2 is relegated to Appendix A.1. The above result implies that the estimates $\mathbf{W}_l$ infinitely often visit a region around the optimal where the norm of the gradient drops below $\frac{1}{\epsilon}\left(\zeta(Q, \delta) + \frac{\delta M}{1 - \delta}\right)$, on average. The size of this near-optimal region depends on the sample complexity of the model $\mathbf{\Phi}$, the Lipschitz constant of the loss function and its gradient, and lastly a design parameter $\epsilon$ of the imposed constraints. The larger $\epsilon$, which is equivalent to imposing an aggressive reduction on the gradients (see (5)), the closer we are guaranteed to get to a local optimal $\mathbf{W}^*$.

**Corollary 1.** *Under the same assumptions of Theorem 2, it holds that*

$$\lim_{l \to \infty} P(\mathbb{E}[\min_{k \le l} \|\nabla f(\mathbf{W}_k)\|] \ge \gamma) \le \frac{1}{\epsilon \gamma} \left( \zeta(Q, \delta) + \frac{\delta M}{1 - \delta} \right). \tag{9}$$

Since Theorem 2 holds on average, we use Markov's inequality to show convergence in probability result in Corollary 1. The size of the near-optimal region $\gamma$ is then controllable by the number of samples $Q$ and the constant $\epsilon$.

In addition to the asymptotic analysis, we aspire to characterize an upper bound of the number of unrolled layers that achieves certain precision. To achieve this, we derive an upper bound to the gradient norm after a finite number of layers $L$ in Theorem 3.

**Theorem 3.** *For a trained unrolled optimizer $\boldsymbol{\theta}^*$ according to Theorem 1, the gradient norm achieved after L layers satisfies*

$$\mathbb{E}\big[\|\nabla f(\mathbf{W}_L)\|\big] \leq (1-\delta)^L (1-\epsilon)^L \, \mathbb{E}\|\nabla f(\mathbf{W}_0)\| + \frac{1}{\epsilon}\left(\zeta(Q,\delta) + \frac{\delta M}{1-\delta}\right).$$

The proof is relegated to Appendix A.3.

## 5   GNN-BASED UNROLLED DGD

SURF is agnostic to the iterative update step $\phi$ that we choose to unroll. However, the standard update rule and the unrolled architecture should accommodate the requirements of the (FL) problem, which are i) to permit distributed execution and ii) to satisfy the consensus constraints of (FL). In this section, we pick DGD as an example of a decentralized algorithm that satisfies the two requirements and unroll it using GNNs, which also can be executed distributedly.

DGD is a distributed iterative algorithm that relies on limited communication between agents. At each iteration $l$, the updating rule of DGD has the form

$$\mathbf{w}_i(l) = \sum_{j \in \mathcal{N}_i} \alpha_{ij} \mathbf{w}_j(l-1) - \beta \nabla f_i(\mathbf{w}_i(l-1)), \quad i = 1, \ldots, n, \tag{10}$$

where $f_i$ is the local objective function, $\beta$ is a fixed step size and $\alpha_{ij} = \alpha_{ji}$ (Nedic and Ozdaglar, 2009, Eq. (3)). The weights $\alpha_{ij}$ are chosen such that $\sum_{j=1}^{n} \alpha_{ij} = 1$ for all $i$ to ensure that (10) converges (Nedic and Ozdaglar, 2009). The update rule in (10) can be interpreted as letting the agents descend in the opposite direction of the local gradient $\nabla f_i(\mathbf{w}_i(l-1))$ as they move away from the (weighted) average of their neighbors' estimates $\mathbf{w}_j(l-1)$. Each iteration can then be divided into two steps; first the agents aggregate information from their direct neighbors and then they calculate the gradient of their local objective functions.

We unfold these two steps in a learnable neural layer of the form

$$\mathbf{w}_{i,l} = [\mathbf{H}_l(\mathbf{W}_{l-1})]_i - \sigma\left(\mathbf{M}_l[\mathbf{w}_{i,l-1}; \mathbf{b}_{i,l}] + \mathbf{d}_l\right), \tag{U-DGD}$$

where $[.]_i$ refers to the $i$-th row of a matrix, $\mathbf{b}_{i,l} = [\mathbf{B}_l]_i$ is the mini-batch used by agent $i$ at layer $l$, and $\sigma$ is a non-linear activation function. In (U-DGD), we replace the first term in (10) with a learnable graph filter (Gama et al., 2020b) and the second term with a single fully-connected layer that is parameterized by $\mathbf{M}_l$ and $\mathbf{d}_l$. The graph filter, the building block of GNNs, aggregates information from up to $(K-1)$-hop neighbors,

$$\mathbf{H}(\mathbf{W}_{l-1}) = \sum_{k=0}^{K-1} h_{k,l} \mathbf{S}^k \mathbf{W}_{l-1}, \tag{11}$$

which, in turn, requires $K-1$ communication rounds. Here, the filter coefficients $\mathbf{h}_l = \{h_{k,l}\}_{k=0}^{K-1}$ that weigh the information aggregated from different hop neighbors are the learnable parameters. Equation (11) and the first term of (10) are essentially the same when $K$ is set to 2 and $h_k$ to 1 for all $k$ and $\mathbf{S}$ is chosen to be the (normalized) graph adjacency matrix. In U-DGD, however, the goal is to learn the weights $h_{k,l}$ to accelerate the unrolled network's convergence.

The other component of (U-DGD) is a single fully-connected perceptron, which is implemented locally and whose weights $\mathbf{M}_l$ and $\mathbf{d}_l$ are shared among all the agents. The input to this perceptron at each agent is the previous estimate $\mathbf{w}_{i,l-1}$ concatenated with a batch of $B$ examples $\mathbf{b}_{i,l}$, sampled randomly from the dataset $\vartheta$. Each batch is a concatenation of the sampled data points, where the input data and label of one example follow each other. Consequently, the size of this fully-connected layer grows fast with the size of the original model $\boldsymbol{\Psi}(.; \mathbf{w})$ that is being collaboratively trained and the size of the mini-batch $\mathbf{b}_{i,l}$. The goal of deploying this fully-connected perceptron is to learn a descending direction over the domain of $f_i$, on average. In our case, $f_i$ is a statistical average of a loss function $\ell$ over the data distribution $\mathcal{D}$.

**Remark 1.** Since the parameters of the fully-connected perceptron are shared between all the agents, U-DGD learners inherit the permutation equivariance of graph filters and graph neural networks, as well as transferability to graphs with different sizes (Ruiz et al., 2020) and stability to small graph perturbations (Gama et al., 2019; 2020a; Hadou et al., 2022; 2023).

# 6 NUMERICAL EXPERIMENTS

In this section, we run experiments relying on U-DGD to showcase the three merits of our method, SURF, in terms of convergence speed, convergence guarantees achieved by the constraints, and robustness against asynchronous communications. More experiments over different graph topologies are relegated to Appendix B including an extension to standard FL scenarios.

**Set-up.** We consider a network of $n = 100$ agents that form a connected 3-degree regular graph. The agents collaborate to train a softmax layer of a classifier via a trained U-DGD optimizer. The softmax layer is fed by the outputs of the convolutional layers of a ResNet18 backbone, whose weights are pre-trained and kept frozen during the training process. To train a U-DGD optimizer via SURF, we consider a meta-training dataset, which consists of 600 class-imbalanced datasets. Each dataset has a different label distribution and contains $6,000$ images (that is, $45$ training examples/agent and $15$ for testing) that are evenly divided between the agents.

**Meta-training.** At each epoch, we randomly choose one image dataset from the meta-training dataset and feed its $45$ training examples/agent to the U-DGD network in mini-batches of $10$ examples/agent at each layer (see (U-DGD)). The training loss is computed over the $10$ testing examples/agent and optimized using ADAM with a learning rate $\mu_\theta = 10^{-2}$ and a dual learning rate $\mu_\lambda = 10^{-2}$. The constraint parameter $\epsilon$ is set to $0.01$. The performance of the trained U-DGD is examined over a meta-testing dataset that consists of 30 class imbalanced datasets, each of which also has $45$ training examples and $15$ for testing per agent. Similar to training, the training examples are fed to the U-DGD in mini-batches while the testing examples are used to compute the testing accuracy. The results are reported for both MNIST (MNISTWebPage) and CIFAR10 (Krizhevsky et al., 2009) datasets. All experiments were run on an NVIDIA® GeForce RTX™ 3090 GPU. Our code is available at: https://anonymous.4open.science/r/fed-SURF-84DC/README.md.

**Convergence speed.** We train a U-DGD that consists of $10$ unrolled layers, each of which employs a graph filter that aggregates information from up to two neighbors (i.e., $K = 3$). That creates a total of $20$ communication rounds between the agents. Figure 1 shows the convergence of the trained U-DGD compared to the standard DGD and DFedAvgM (Sun et al., 2023). The former takes only $20$ communication rounds to achieve performance higher than that achieved by the others in $200$ communication rounds. This result is confirmed for both MNIST (left) and CIFAR10 (Middle). In addition, we compute the relative accuracy with respect to centralized training and report the result in Figure 1 (Right). The figure confirms that U-DGD achieves a relative accuracy of almost $1$, which indicates that SURF matches the performance of centralized training. The figure also shows a comparison with other federated learning benchmarks: FedAvg (McMahan et al., 2016), SCAFFOLD (Karimireddy et al., 2020), MOON (Li et al., 2021), FedProx (Li et al., 2020a), and FedDyn (Acar et al., 2021). U-DGD has a notably faster convergence than all the benchmarks.

**Convergence guarantees.** To assess the effects of the descending constraints on the training performance, we compare the test loss and accuracy with and without these constraints in Figure 2. The figure shows that the unrolled optimizer trained using SURF, depicted in blue, converges gradually to the optimal loss/accuracy over the layers. However, the standard unrolled optimizer trained without the descending constraints failed to maintain a similar behavior even though it achieves the same

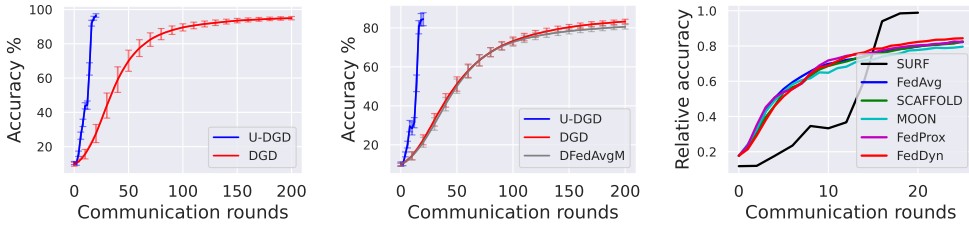

Figure 1: **Convergence speed.** Comparisons between the accuracy of U-DGD and DGD evaluated over 30 in-distribution class-imbalanced datasets sampled from (Left) MNIST and (Middle) CIFAR10. (Right) Comparison of the relative accuracy (i.e., the accuracy normalized by that of centralized training) over CIFAR10 between SURF and FL benchmarks that deploy a central server.

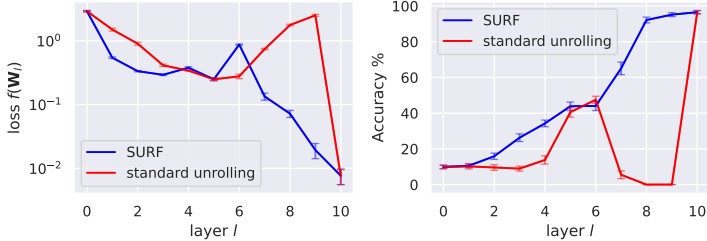

Figure 2: **Convergence Guarantees.** Comparison of the loss and accuracy (evaluated over 30 test datasets sampled from MNIST) with and without the constraints in (SURF) across the unrolled layers of U-DGD. Observe that SURF converges gradually to the optimal.

performance at the final layer. In fact, the accuracy jumps from $0\%$ to $96\%$ at the last layer, which would make the optimizer more vulnerable to additive noise and perturbations in the layers' inputs, as we show in the following experiment.

**Asynchronous communications.** We consider an asynchronous setting during inference, where at each layer $n_{asyn}$ randomly-chosen agents are asynchronous with the rest of the agents. The asynchronous agents fail to update their estimates simultaneously with the other agents. Relying on outdated estimates introduces perturbations to the inputs of each layer, which would result in a discrepancy in the learned descent direction. Figure 3 shows that our constrained method SURF is more resilient to these perturbations, as the deterioration in the performance is notably slower than that of the case with no constraints.

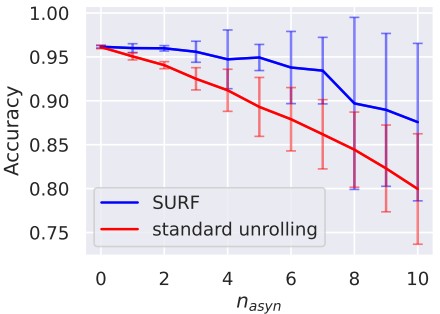

Figure 3: **Asynchronous Communications.** Comparison of the test loss and accuracy in different communication environments where $n_{asyn}$ agents are asynchronous with the rest of the network.

## 7 CONCLUSIONS

In this paper, we proposed a new framework, called SURF, that introduces stochastic algorithm unrolling to federated learning scenarios. To overcome the brittleness of algorithm unrolling, SURF imposes descending constraints on the outputs of the unrolled layers. These constraints provided our method with resilience to the perturbations induced by both feeding the unrolled layers with stochastic mini-batches and asynchronous communications. SURF, however, is independent of the standard algorithm to be unrolled. For the federated-learning scenario we considered, in this paper, we unrolled DGD using GNNs, which allowed distributed execution of the optimizer along with transferability to different regimes. Our analysis showed that the unrolled DGD almost-surely converges to a near-optimal region whose size depends on the sample complexity of the unrolled network, the smoothness of the loss function, and a design parameter of the descending constraints.

There are several directions for future work. One possible direction is to expand our method to more challenging federated learning scenarios. Although our work assumed homogeneity among the agents, there is ample opportunity to extend our method to heterogeneous settings through loss reweighting techniques such as (Zhao and Joshi, 2022). Moreover, privacy is a critical concern in federated learning, since even though the agents do not share their data, they communicate their evaluated gradients, which can be exploited in inferring the data. Unrolled optimizers are prone to the same privacy issues since the input of the fully-connected perceptron can be inferred from its outputs (Fredrikson et al., 2015). Methods inspired by differential privacy (Abadi et al., 2016; Arachchige et al., 2019) and secure aggregation (So et al., 2021; Elkordy et al., 2022) can be further explored in the context of stochastic unrolling. Another direction of interest is to explore the use of our method in other learning paradigms, namely reinforcement learning and neural ODEs (Chen et al., 2018a).

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

# A  PROOFS

In this section, we provide the proofs for our theoretical results after introducing the following notation. Consider a probability space $(\Omega, \mathcal{F}, P)$, where $\Omega$ is a sample space, $\mathcal{F}$ is a sigma algebra, and $P : \mathcal{F} \to [0, 1]$ is a probability measure. With a slight abuse of this measure-theoretic notation, we write $P(X = 0)$ instead of $P(\{\omega : X(\omega) = 0\})$, where $X : \Omega \to \mathbb{R}$ is a random variable, to keep equations concise. We define a filtration of $\mathcal{F}$ as $\{\mathcal{F}_l\}_{l>0}$, which can be thought of as an increasing sequence of $\sigma$-algebras with $\mathcal{F}_{l-1} \subset \mathcal{F}_l$. We assume that the outputs of the unrolled layers $\mathbf{W}_l$ are adapted to $\mathcal{F}_l$, i.e., $\mathbf{W}_l \in \mathcal{F}_l$. Intuitively, the filtration $\mathcal{F}_l$ describes the information at our disposal at step $l$, which includes the outputs of each layer up to layer $l$, along with the initial estimate $\mathbf{W}_0$.

In our proofs, we use a supermartingale argument, which is commonly used to prove the convergence of stochastic descent algorithms. A stochastic process $X_k$ is said to form a supermartingale if $\mathbb{E}[X_k | X_{k-1}, \ldots, X_0] \leq X_{k-1}$. This inequality implies that given the past history of the process, the future value $X_k$ is not, on average, larger than the latest one. With this definition in mind, we provide the proof of Theorem 2.

## A.1  PROOF OF THEOREM 2

Let $A_l \in \mathcal{F}_l$ be the event that the constraint (5) at layer $l$ is satisfied. By the law of total expectation, we have

$$\mathbb{E}\big[\|\nabla f(\mathbf{W}_l)\|\,\big] = P(A_l)\mathbb{E}\Big[\|\nabla f(\mathbf{W}_l)\| \,|A_l\Big] + P(A_l^c)\mathbb{E}\Big[\|\nabla f(\mathbf{W}_l)\| \,|A_l^c\Big], \qquad (12)$$

with $P(A_l) = 1 - \delta$. On the right-hand side, the first term represents the conditional expectation when the constraint is satisfied and, in turn, is bounded above according to (5). The second term is concerned with the complementary event $A_l^c \in \mathcal{F}_l$, when the constraint is violated. The conditional expectation in this case can also be bounded since i) the gradient norm $\|\nabla f(\mathbf{W}_l)\| \leq M$ for all $\mathbf{W}_l$ since $f$ is $M$-Lipschitz according to Assumption 1, and ii) the expectation of a random variable cannot exceed its maximum value, i.e, $\mathbb{E}\|\nabla f(\mathbf{W}_l)\| \leq \max_{\mathbf{W}_l} \|\nabla f(\mathbf{W}_l)\| \leq M$ by Cauchy-Schwarz inequality. Substituting in (12) results in an upper bound of

$$\mathbb{E}\big[\|\nabla f(\mathbf{W}_l)\|\big] \leq (1-\delta)(1-\epsilon)\,\mathbb{E}\|\nabla f(\mathbf{W}_{l-1})\| + (1-\delta)\zeta(Q,\delta) + \delta M, \qquad (13)$$

almost surely.

In the rest of the proof, we leverage the supermartingale convergence theorem to show that (13) indeed implies the required convergence. We start by defining a sequence of random variables $\{Z_l\}_l$ each of which has a degenerative distribution such that

$$Z_l = \mathbb{E}\|\nabla f(\mathbf{W}_l)\| \quad a.s. \quad \forall l. \qquad (14)$$

Then, we form a supermartingale-like inequality using the law of total expectation. That is, we have

$$\mathbb{E}[Z_l \,|\, \mathcal{F}_{l-1}] \leq (1-\delta)(1-\epsilon)\,Z_{l-1} + (1-\delta)\zeta(Q,\delta) + \delta M$$
$$= (1-\delta)\,Z_{l-1} - (1-\delta)\Big(\epsilon Z_{l-1} - \zeta(Q,\delta) - \frac{\delta M}{1-\delta}\Big). \qquad (15)$$

The structure of the proof is then divided into two steps. First, we prove that when $l$ grows, $Z_l$ almost surely and infinitely often achieves values below $\frac{1}{\epsilon}\big(\zeta(Q,\delta) + \delta M/1-\delta\big)$. Second, we show that this is also true for the gradient norm $\|\nabla f(\mathbf{W}_l)\|$ itself. This implies that the outputs of the unrolled layers enter a near-optimal region infinitely often.

To tackle the first objective, we define the lowest gradient norm achieved, on average, up to layer $l$ as $Z_l^{\text{best}} = \min_{k \leq l}\{Z_k\}$. To ensure that $Z_l$ enters this region infinitely often, it suffices to show that

$$\lim_{l \to \infty} Z_l^{\text{best}} \leq \frac{1}{\epsilon}\big(\zeta(Q,\delta) + \delta M/1-\delta\big) \quad a.s. \qquad (16)$$

To show that the above inequality is true, we start by defining the sequences

$$\alpha_l := Z_l \cdot \mathbf{1}\Big\{Z_l^{\text{best}} > \frac{1}{\epsilon}\big(\zeta(Q,\delta) + \delta M/1-\delta\big)\Big\},$$
$$\beta_l := \Big(\epsilon Z_l - \zeta(Q,\delta) - \frac{\delta M}{1-\delta}\Big)\mathbf{1}\Big\{Z_l^{\text{best}} > \frac{1}{\epsilon}\big(\zeta(Q,\delta) + \delta M/1-\delta\big)\Big\}, \qquad (17)$$

where $\mathbf{1}\{.\}$ is an indicator function. The first sequence $\alpha_l$ tracks the values of $Z_l$ until the best value $Z_l^{\text{best}}$ drops below the threshold $\frac{1}{\epsilon}\left(\zeta(Q,\delta)+\delta M/1-\delta\right)$ for the first time. After this point, the best value stays below the threshold since $Z_{l+1}^{\text{best}} \leq Z_l^{\text{best}}$ by definition, which implies that the indicator function stays zero and $\alpha_l = 0$. In other words, we have

$$\alpha_l = \left\{ \begin{array}{cc} Z_l & l < T \\ 0 & \text{otherwise,} \end{array} \right. \tag{18}$$

with $T := \min\{l \mid Z_l^{\text{best}} \leq \frac{1}{\epsilon}\left(\zeta(Q,\delta)+\delta M/1-\delta\right)\}$. Similarly, the sequence $\beta_l$ follows the values of $\epsilon Z_l - \zeta(Q,\delta) - \frac{\delta}{1-\delta}M$ until it falls below zero for the first time, which implies that $\beta_k \geq 0$ by construction. Moreover, it also holds that $\alpha_l \geq 0$ for all $l$ since $Z_l$ is always non-negative.

We now aim to show that $\alpha_l$ forms a supermartingale, so we can use the supermartingale convergence theorem to prove (16). This requires finding an upper bound of the conditional expectation $\mathbb{E}[\alpha_l|\mathcal{F}_{l-1}]$. We separate this expectation into two cases, $\alpha_{l-1} = 0$ and $\alpha_{l-1} \neq 0$, and use the law of total expectation to write

$$\mathbb{E}[\alpha_l|\mathcal{F}_{l-1}] = \mathbb{E}[\alpha_l|\mathcal{F}_{l-1},\alpha_{l-1}=0]P(\alpha_{l-1}=0) + \mathbb{E}[\alpha_l|\mathcal{F}_{l-1},\alpha_{l-1}\neq 0]P(\alpha_{l-1}\neq 0). \tag{19}$$

First, we focus on the case when $\alpha_{l-1} = 0$, and for conciseness, let $\eta := \frac{1}{\epsilon}\left(\zeta(Q,\delta)+\delta M/(1-\delta)\right)$ be the radius of the near-optimal region centered around the optimal. Equation (17) then implies that the indicator function is zero, i.e., $Z_l^{\text{best}} \leq \eta$, since the non-negative random variable $Z_l$ cannot be zero without $Z_l^{\text{best}} \leq \eta$. It also follows that $\beta_{l-1}$ is zero since it employs the same indicator function as $\alpha_l$. As we discussed earlier, once $\alpha_{l-1} = 0$, all the values that follow are also zero, i.e., $\alpha_k = 0, \forall k \geq l-1$ (c.f. (18)). Hence, the conditional expectation of $\alpha_l$ can be written as

$$\mathbb{E}[\alpha_l|\mathcal{F}_{l-1},\alpha_{l-1}=0] = 0 =: (1-\delta)(\alpha_{l-1}-\beta_{l-1}). \tag{20}$$

On the other hand, when $\alpha_{l-1} \neq 0$, the conditional expectation follows from the definition in (17),

$$\begin{aligned} \mathbb{E}[\alpha_l|\mathcal{F}_{l-1},\alpha_{l-1}\neq 0] &= \mathbb{E}[Z_l \cdot \mathbf{1}\{Z_l^{\text{best}} > \eta\}|\mathcal{F}_{l-1},\alpha_{l-1}\neq 0] \\ &\leq \mathbb{E}[Z_l|\mathcal{F}_{l-1},\alpha_{l-1}\neq 0] \\ &\leq (1-\delta)Z_{l-1} - (1-\delta)\left(\epsilon Z_{l-1} - \zeta(Q,\delta) - \frac{\delta M}{1-\delta}\right) \\ &= (1-\delta)(\alpha_{l-1}-\beta_{l-1}). \end{aligned} \tag{21}$$

The first inequality is true since the indicator function is at most one, and the second inequality is a direct application of (15). The last equality results from the fact that the indicator function $\mathbf{1}\{Z_l^{\text{best}} > \eta\}$ is 1 since $\alpha_{l-1} \neq 0$, which implies that $\alpha_{l-1} = Z_{l-1}$ and $\beta_{l-1} = \epsilon Z_{l-1} - \zeta(Q,\delta) - \frac{\delta}{1-\delta}M$. Combining the results in (20) and (21) and substituting in (19), it finally follows that

$$\begin{aligned} \mathbb{E}[\alpha_l|\mathcal{F}_{l-1}] &\leq (1-\delta)(\alpha_{l-1}-\beta_{l-1})[P(\alpha_{l-1}=0)+P(\alpha_{l-1}\neq 0)] \\ &= (1-\delta)(\alpha_{l-1}-\beta_{l-1}), \end{aligned} \tag{22}$$

and we emphasize that both $\alpha_{l-1}$ and $\beta_{l-1}$ are non-negative by definition.

Given (22), it follows from supermartingale convergence theorem (Robbins and Siegmund, 1971, Theorem 1) that (i) $\alpha_l$ converges almost surely, and (ii) $\sum_{i=1}^{\infty} \beta_l$ is almost surely summable (i.e., finite). When the latter is written explicitly, we get

$$\sum_{l=1}^{\infty}\left(\epsilon Z_l - \zeta(Q,\delta) - \frac{\delta M}{1-\delta}\right)\mathbf{1}\{Z_l^{\text{best}} > \eta\} < \infty, \quad a.s., \tag{23}$$

The almost sure convergence of the above sequence implies that the limit inferior and limit superior coincide and

$$\liminf_{l\to\infty}\left(\epsilon Z_l - \zeta(Q,\delta) - \frac{\delta M}{1-\delta}\right)\mathbf{1}\{Z_l^{\text{best}} > \eta\} = 0, \quad a.s. \tag{24}$$

The latter is true if either there exist a sufficiently large $l$ such that $Z_l^{\text{best}} \leq \eta = \frac{1}{\epsilon}\left(\zeta(Q,\delta)+\delta M/1-\delta\right)$ or it holds that

$$\liminf_{l\to\infty}\left(\epsilon Z_l - \zeta(Q,\delta) - \frac{\delta M}{1-\delta}\right) = 0, \quad a.s. \tag{25}$$

The above equation can be re-written as $\sup_l \inf_{m \geq l} Z_m = \frac{1}{\epsilon}\left(\zeta(Q,\delta) + \frac{\delta M}{1-\delta}\right)$. Hence, there exists some large $l$ where $Z_l^{\text{best}} \leq \sup_l \inf_{m \geq l} Z_m$, which again reaches the same conclusion. This proves the correctness of (16).

To this end, we have shown the convergence of $Z_l^{\text{best}}$, which was defined as the best *expected* value of $\|\nabla f(\mathbf{W}_l)\|$. It is still left to show the convergence of the random variable $\|\nabla f(\mathbf{W}_l)\|$ itself. Start with writing $Z_l = \int \|\nabla f(\mathbf{W}_l)\| dP$, which turns (25) into

$$\liminf_{l \to \infty} \int \epsilon \|\nabla f(\mathbf{W}_l)\| dP = \zeta(Q,\delta) + \frac{\delta M}{1-\delta}, \quad a.s. \tag{26}$$

By Fatou's lemma (Durrett, 2019, Theorem 1.5.5), it follows that

$$\int \liminf_{l \to \infty} \epsilon \|\nabla f(\mathbf{W}_l)\| dP \leq \liminf_{l \to \infty} \int \epsilon \|\nabla f(\mathbf{W}_l)\| dP = \zeta(Q,\delta) + \frac{\delta M}{1-\delta}. \tag{27}$$

We can bound the left hand side from below by defining $f_l^{\text{best}} := \min_{k \leq l} \|\nabla f(\mathbf{W}_k)\|$ as the lowest gradient norm achieved up to layer $l$. By definition, $f_l^{\text{best}} \leq \liminf_{l \to \infty} \|\nabla f(\mathbf{W}_l)\|$ for sufficiently large $l$. Therefore, we get

$$\epsilon \int f_l^{\text{best}} dP \leq \epsilon \int \liminf_{l \to \infty} \epsilon \|\nabla f(\mathbf{W}_l)\| dP \leq \zeta(Q,\delta) + \frac{\delta M}{1-\delta}, \quad a.s. \tag{28}$$

for some large $l$. Equivalently, we can write that

$$\lim_{l \to \infty} \int f_l^{\text{best}} dP \leq \frac{1}{\epsilon}\left(\zeta(Q,\delta) + \frac{\delta M}{1-\delta}\right), \quad a.s. \tag{29}$$

which completes the proof.

## A.2 PROOF OF COROLLARY 1

Using Markov's inequality, it follows from (9) that

$$\lim_{l \to \infty} P(|f_l^{\text{best}}| \geq \gamma) \leq \lim_{l \to \infty} \frac{\mathbb{E} f_l^{\text{best}}}{\gamma} \leq \frac{1}{\epsilon \gamma}\left(\zeta(Q,\delta) + \frac{\delta M}{1-\delta}\right), \tag{30}$$

where we drop the absolute value in the middle term since $f_l^{\text{best}}$ is almost surely non-negative.

## A.3 PROOF OF THEOREM 3

*Proof.* First, we recursively unroll the right-hand side of (13) to evaluate the reduction in the gradient norm $\mathbb{E}\|\nabla f(\mathbf{W}_l)\|$ after $l$ layers. This leads to the inequality

$$\mathbb{E}\left[\|\nabla f(\mathbf{W}_l)\|\right] \leq (1-\delta)^l (1-\epsilon)^l \, \mathbb{E}\|\nabla f(\mathbf{W}_0)\|$$
$$+ \sum_{i=0}^{l-1}(1-\delta)^{i-1}(1-\epsilon)^{i-1}\Big[(1-\delta)\zeta(Q,\delta) + \delta M\Big]. \tag{31}$$

The right-hand side can be further simplified by evaluating the geometric sum resulting in

$$\mathbb{E}\left[\|\nabla f(\mathbf{W}_l)\|\right] \leq (1-\delta)^l (1-\epsilon)^l \, \mathbb{E}\|\nabla f(\mathbf{W}_0)\|$$
$$+ \frac{1-(1-\delta)^l(1-\epsilon)^l}{1-(1-\delta)(1-\epsilon)}\Big[(1-\delta)\zeta(Q,\delta) + \delta M\Big]. \tag{32}$$

Second, we evaluate the distance between $\mathbb{E}\|\nabla f(\mathbf{W}_L)\|$ at the $L$-th layer and its optimal value

$$\left|\mathbb{E}\left[\|\nabla f(\mathbf{W}_L)\|\right] - \mathbb{E}\left[\|\nabla f(\mathbf{W}^*)\|\right]\right|$$
$$= \lim_{l \to \infty}\left|\mathbb{E}\left[\|\nabla f(\mathbf{W}_L)\|\right] - \mathbb{E}[\min_{k \leq l}\|\nabla f(\mathbf{W}_k)\|] + \mathbb{E}[\min_{k \leq l}\|\nabla f(\mathbf{W}_k)\|] - \mathbb{E}\left[\|\nabla f(\mathbf{W}^*)\|\right]\right|. \tag{33}$$

We add and subtract $\mathbb{E}[\min_{k \leq l} \|\nabla f(\mathbf{W}_k)\|]$ in the right-hand side while imposing the limit when $l$ goes to infinity so we can use triangle inequality. We, hence, get

$$
\begin{aligned}
\left| \mathbb{E}\big[\|\nabla f(\mathbf{W}_L)\|\big] - \mathbb{E}\big[\|\nabla f(\mathbf{W}^*)\|\big] \right| & \\
\leq \lim_{l \to \infty} &\left| \mathbb{E}\big[\|\nabla f(\mathbf{W}_L)\|\big] - \mathbb{E}[\min_{k \leq l} \|\nabla f(\mathbf{W}_k)\|] \right| \\
+ \lim_{l \to \infty} &\left| \mathbb{E}[\min_{k \leq l} \|\nabla f(\mathbf{W}_k)\|] - \mathbb{E}\big[\|\nabla f(\mathbf{W}^*)\|\big] \right|.
\end{aligned}
\tag{34}
$$

Note that the gradient of $f$ at the stationary point $\mathbf{W}^*$ is zero. Therefore, the second term on the right-hand side is upper bounded according to Theorem 2.

The final step required to prove Theorem 3 is to evaluate the first term in (37). To do so, we observe that

$$
\lim_{l \to \infty} \left| \mathbb{E}\big[\|\nabla f(\mathbf{W}_L)\|\big] - \mathbb{E}[\min_{k \leq l} \|\nabla f(\mathbf{W}_k)\|] \right| = \mathbb{E}\big[\|\nabla f(\mathbf{W}_L)\|\big] - \lim_{l \to \infty} \mathbb{E}[\min_{k \leq l} \|\nabla f(\mathbf{W}_k)\|].
\tag{35}
$$

This is the case since $\mathbb{E}\big[\|\nabla f(\mathbf{W}_L)\|\big]$ cannot go below the minimum of the gradient norm when $l$ goes to infinity. Therefore, we can using (32)

$$
\begin{aligned}
\lim_{l \to \infty} &\left| \mathbb{E}\big[\|\nabla f(\mathbf{W}_L)\|\big] - \mathbb{E}[\min_{k \leq l} \|\nabla f(\mathbf{W}_k)\|] \right| \\
&= (1-\delta)^L (1-\epsilon)^L \, \mathbb{E}\|\nabla f(\mathbf{W}_0)\| + \frac{1 - (1-\delta)^L (1-\epsilon)^L}{1 - (1-\delta)(1-\epsilon)} \Big[ (1-\delta)\zeta(Q,\delta) + \delta M \Big] \\
&\quad - \lim_{l \to \infty} (1-\delta)^l (1-\epsilon)^l \, \mathbb{E}\|\nabla f(\mathbf{W}_0)\| - \lim_{l \to \infty} \frac{1 - (1-\delta)^l (1-\epsilon)^l}{1 - (1-\delta)(1-\epsilon)} \Big[ (1-\delta)\zeta(Q,\delta) + \delta M \Big] \\
&= (1-\delta)^L (1-\epsilon)^L \, \mathbb{E}\|\nabla f(\mathbf{W}_0)\| - \frac{(1-\delta)^L (1-\epsilon)^L}{1 - (1-\delta)(1-\epsilon)} \Big[ (1-\delta)\zeta(Q,\delta) + \delta M \Big] \\
&\leq (1-\delta)^L (1-\epsilon)^L \, \mathbb{E}\|\nabla f(\mathbf{W}_0)\|.
\end{aligned}
\tag{36}
$$

Note that the first limit in (36) goes to zero and the second limit is evaluated as the constant $(1-\delta)\zeta(Q,\delta) + \delta M$ divided by $1 - (1-\delta)(1-\epsilon)$. The final inequality

Combining the two results, we can bound the quantity in (37) as follows;

$$
\begin{aligned}
\left| \mathbb{E}\big[\|\nabla f(\mathbf{W}_L)\|\big] - \mathbb{E}\big[\|\nabla f(\mathbf{W}^*)\|\big] \right| & \\
\leq (1-\delta)^L (1-\epsilon)^L \, \mathbb{E}\|\nabla f(\mathbf{W}_0)\| &+ \frac{1}{\epsilon} \left( \zeta(Q,\delta) + \frac{\delta M}{1-\delta} \right),
\end{aligned}
\tag{37}
$$

which completes the proof. $\qquad\square$

Theorem 3 can easily give an upper bound of the number of unrolled layers $L$ needed to achieve certain precision $\gamma$. That is,

$$
L \leq \frac{\log\left( \gamma - \frac{1}{\epsilon}\zeta(Q,\delta) + \frac{\delta M}{\epsilon(1-\delta)} \right) - \log(\mathbb{E}\|\nabla f(\mathbf{W}_0)\|)}{\log(1-\delta) + \log(1-\epsilon)},
\tag{38}
$$

which depends on the sample complexity $\zeta(Q,\delta)$, the Lipschitz constant $M$, and the constraint parameter $\epsilon$.

## B   EXTENDED EXPERIMENTS

In this section, we provide extended experiments with different graph topologies and heterogeneous setups.

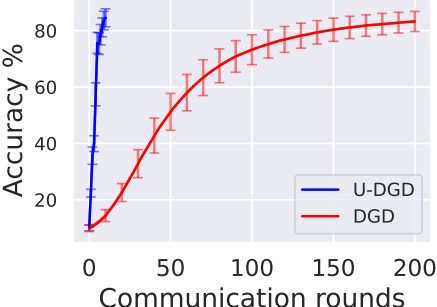 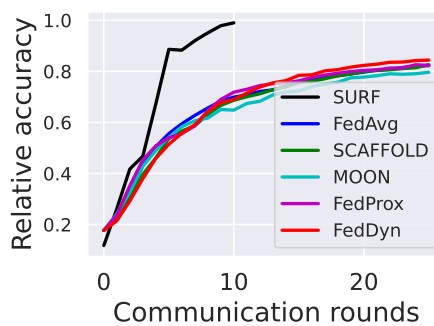

Figure 4: **Standard FL with star graphs.** (Left) Comparisons between convergence rates of U-DGD and DGD evaluated over CIFAR10. (Right) Comparison of the relative accuracy (i.e., the accuracy normalized by that of centralized training) over CIFAR10 between SURF and FL benchmarks that deploy a central server.

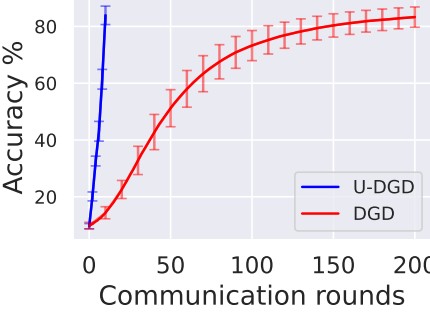 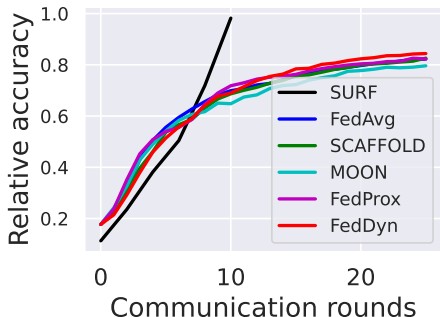

Figure 5: **Serverless FL with random graphs.** (Left) Comparisons between convergence rates of U-DGD and DGD evaluated over CIFAR10. (Right) Comparison of the relative accuracy (i.e., the accuracy normalized by that of centralized training) over CIFAR10 between SURF and FL benchmarks that deploy a central server.

## B.1 STANDARD FL VIA STAR GRAPHS

Our method, SURF, is not restricted to serverless FL. In fact, SURF considers a more general case that can be altered to fit standard FL scenarios with central servers by choosing the underlying graph of the network to be a star graph. Star graphs have $n - 1$ nodes with node degree of $1$ and one central node with node degree of $n - 1$. That is, each node in the graph is only connected to one central node, which serves as a central server. We repeat our experiments under this new scenario. We set $\mu_\theta = 10^{-3}$, $\epsilon = 0.1$, and $K = 2$ while the rest of the parameters are kept the same as they were in the main experiment in Section 6. In this case, setting the filter taps $K$ to 2 implies that each node communicate only with their 1-hop neighbors and therefore only one communication round is required at each layer. Figure 4 shows the convergence rates of U-DGD trained with SURF compared to DGD and other standard FL benchmarks. The figure suggests that SURF achieves the same performance attained by central training in 10 communication rounds while the other benchmarks need 25 rounds to reach almost $80\%$ of the centralized performance. This is aligned with the results in Figure 1, which indicates that the fast convergence does not rely on the graph topology.

## B.2 RANDOM GRAPHS

We also repeat our experiments for random graphs, where an edge is drawn between two nodes on the graph with probability $p$. The experiment parameters are kept the same as the original experiments except that we use a U-DGD model with 5 layers. In our experiments, we set $p$ to 0.1 and $\epsilon$

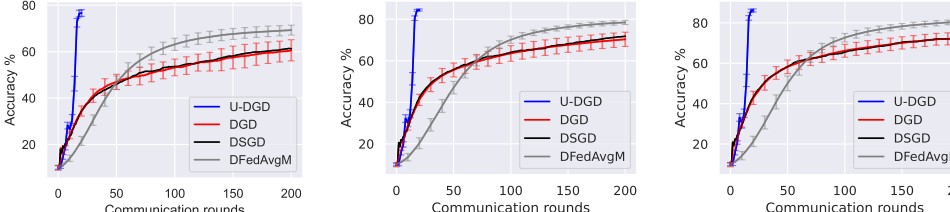

Figure 6: **Heterogeneous settings.** Comparisons between the accuracy of U-DGD and other decentralized benchmarks evaluated over 30 class-imbalanced CIFAR10 datasets sampled according to a Dirichlet distribution with a concentration parameter (Left) $\alpha = 0.3$, (Middle) $\alpha = 0.7$, and (Right) $\alpha = 1$, The higher $\alpha$, the less heterogeneous the agents are. U-DGD is more robust than the other benchmarks.

to $0.1$. The convergence rate is reported in Figure 5, which shows a similar behavior to the previous experiments. Thus, we conclude that the graph togology does not contribute to the fast convergence of SURF. The fast convergence is a result of learning from training data a sequence of descending steps that converges fast.

### B.3    HETEROGENEOUS SETTINGS

We test our unrolled model, U-DGD, on heterogeneous agents who sample their data according to a Dirichlet distribution with a concentration parameter $\alpha$. The lower $\alpha$, the more heterogeneous the agents are. In Figure 6, we compare the accuracy of U-DGD to other decentralized FL benchmarks: DGD (c.f. (10)), distributed stochastic gradient descent (DSGD), and decentralized federated averaging (DFedAvgM) (Sun et al., 2023). In both DGD and DSGD, the agents update their estimate based on their local data through one gradient step at each communication round. The gradients in DGD are computed over a mini-batch of 10 data points/agent compared to one data point in DSGD. In DFedAvgM, each agent takes 6 gradient steps with momentum at each communication round. U-DGD therefore has the lowest update rate as it occurs once at each two communication rounds. Figure 6 then shows that U-DGD is more robust to the agent heterogeneity while the other benchmarks are more prone to client drift. It is clear that the performance deteriorates when $\alpha$ decreases over all the methods. However, the deterioration in U-DGD's performance is much slower compared to the other methods. This is the case since U-DGD, during its meta training, learns to converge faster while balancing between the the aggregated models received from the agents' neighbors and the local updates.

