# OpenReview forum: "Stochastic Unrolled Federated Learning"
_ICLR.cc/2024/Conference — Submitted to ICLR 2024_

### Official Review · Reviewer_DuLq · 2023-10-27

**Soundness:** 3 good
**Presentation:** 3 good
**Contribution:** 2 fair
**Rating:** 6
**Confidence:** 4

**Summary:**

The paper introduces Stochastic UnRolled Federated learning (SURF), a novel approach that applies algorithm unrolling, a learning-based optimization paradigm, to the server-free federated learning scenario. The authors aim to leverage these benefits to address the challenges faced by low-end devices in collaborative deep model training. The paper identifies two main challenges in applying algorithm unrolling to federated learning: the necessity of feeding whole datasets to unrolled optimizers and the decentralized nature of federated learning. The authors propose solutions to these challenges by introducing stochastic mini-batches and a graph neural network (GNN)-based unrolled architecture, respectively. The stochastic mini-batches address the data feeding issue, while the GNN-based architecture preserves the decentralized nature of federated learning. The authors also provide theoretical proof of the convergence of their proposed unrolled optimizer and demonstrate its efficacy through numerical experiments.

**Strengths:**

1. Originality: The paper introduces a novel approach. Algorithm unrolling is a learning-based optimization paradigm where iterative algorithms are unfolded into trainable neural networks, leading to faster convergence. Federated learning, on the other hand, is a distributed learning paradigm where multiple devices collaboratively train a global model. The originality of the paper lies in its integration of these two concepts, addressing specific challenges in server-free federated learning such as the need for whole datasets in unrolled optimizers and the decentralized nature of the learning process.

2. Clarity: The paper is well-structured and presents its ideas in a clear and concise manner.

3. Algorithm Simplicity and Neatness: Despite addressing complex challenges in federated learning, the algorithm proposed in the paper is simple and neat. The use of stochastic mini-batches and a GNN-based architecture provides a straightforward yet effective solution. The simplicity of the algorithm makes it accessible and easy to implement.

**Weaknesses:**

1. Vulnerability of Assumption 1: The paper assumes convexity in its problem formulation, which might not align with the real-world scenarios where deep learning models, predominantly used in Federated Learning (FL), are non-convex. This assumption is quite vulnerable as it oversimplifies the complexity of the learning models, potentially leading to over-optimistic results and conclusions. In practice, dealing with non-convex optimization problems is more challenging, and the algorithms need to be robust enough to handle such complexities.

2. Practicality of Assumption 2: The assumption that  g=f and g=∣∣∇f∣∣ (f=∣∣∇f∣∣) is very rare to satisfy in real-world applications. These conditions impose strict requirements on the relationship.

3. Local Minima and Convergence: In non-convex optimization problems, the paper should consider replacing the goal of reaching local minima with finding stationary points, which are points where the gradient is close to zero. This adjustment would provide a more accurate representation of the convergence behavior in non-convex settings, since two neural nets are involved.

4. Heterogeneity of Local Models and Fair Comparison: The paper adopts the heterogeneity of local models and data distribution in federated learning settings. However, the comparison of SURF with FedAvg-type methods might not be entirely fair due to this heterogeneity. To address this issue, the paper should conduct more extensive experiments, comparing SURF with a broader range of personalized federated learning methods that are designed to handle heterogeneity more effectively. Some of the methods that could be considered for comparison:

pFedMe: Personalized Federated Learning with Moreau Envelopes Dinh et al., 2020
PerFedAvg: Personalized Federated Learning with Theoretical Guarantees: A Model-Agnostic Meta-Learning Approach Fallah et al., 2020
APFL: Adaptive Personalized Federated Learning Deng et al., 2020
Ditto: Fair and Robust Federated Learning Through Personalization Li et al., 2022
Mobilizing Personalized Federated Learning in Infrastructure-Less and Heterogeneous Environments via Random Walk Stochastic ADMM
, Parsons et al., 2023

**Questions:**

1. Given that the assumption of convexity might not hold in many real-world deep learning scenarios, how does this affect the applicability of SURF, and are there plans to extend SURF to non-convex settings?
2. How can we ensure that the conditions g=f and g=∣∣∇f∣∣ are met?
3. How much does the heterogeneity of local models and data distribution in federated learning environments affect the performance of SURF?

---

> ### Author Response · Authors · 2023-11-20
>
> We would like to thank the reviewer for their insightful review, and for their positive evaluation of our work. We would like to address the reviewer's comments and questions combined as follows:
>
> 1. We thank the reviewer for their deliberate reading of our paper and theoretical analysis. The reviewer is indeed correct about the fact that the convexity assumption does not hold in practice. We acknowledge that we made an unintentional mistake in our write-up. Assumptions 1-5 are inherited from the constrained learning theory (Theorem 1 in [1]). This theorem requires strong duality of a version of (SURF) that learns a continuous function $\Phi$ instead of the parameterized one. The strong duality holds if $f$ is convex. Another sufficient conditions for the strong duality to hold is by satisfying Assumptions 3 and 5, as proved in Proposition 3.2 in [1]. In our case, $f$ is definitely non-convex and therefore we had Assumptions 3 and 5 in place. These two assumptions are mild and hold for modern DL models.  We have corrected the mistake in Assumption 1 in our revised manuscript and we thank the reviewer again for their deliberate reading of our paper.
> 3. Assumption 2 identifies the sample complexity required to approximate the actual expectations with sample averages in both the objective function and the constraints of (SURF). Therefore, Assumption 2 only suggests that we need $Q$ realizations to approximate both $\mathbb{E}[f]$ and $\mathbb{E}[|\nabla f|]$ with at least $\zeta$ precision where $\zeta$ monotonically decreases with $Q$. This is a typical assumption in deep learning. **We, therefore, do not require $f = \|\nabla f\|$**. We meant that the sample complexity holds for the two cases. To prevent this confusion, we split this assumption into two parts, each of which handles one case separately.
> 4. We agree with the reviewer. The reviewer's suggestion is exactly what we propose in our paper. We force the unrolled network to converge to a stationary point by imposing the descending constraints in (SURF). These constraints force the gradients to decrease over the layers. We acknowledge that the notation in Theorem 2 would imply otherwise. We have adjusted our notation in the revised manuscript to make it clear that we converge to a region where the gradient of $f$ is upper bounded by a small value that is close to zero. This definitely implies that we converged to a stationary point.
> 5. While related, our method does not directly belong to the personalized FL class of algorithms. The (FL) problem we solve imposes consensus constraints between the agents. These constraints force all the agents to learn the same model despite the disparity between the agents. DGD was proved in [2] to converge to a solution where all $w_i$'s coincide. Unrolled DGD that we propose is an accelerated version of DGD that can converge to the same solution in less iterations. Therefore, without additional changes, we do not expect our method to be able to compete with personalized FL methods over performance measures. We emphasize that the advantage of our method is in the fast convergence and not the performance. Indeed, The FL baselines depicted in Figure 1 will reach the same performance level of SURF eventually when we run more iterations. Therefore, we believe that comparisons with personalized FL is not fair to our method. Having said that, we agree that heterogeneity is a challenging  issue in FL that we aim to consider in a future work.
>
> [1] L. Chamon et al., Constrained learning with non-convex losses, IEEE Transactions on Information Theory, 2022.
> [2] A. Nedic et al., Distributed subgradient methods for multi-agent optimization, IEEE Transactions on Automatic Control, 54(1):48–61, 2009.

---

### Official Review · Reviewer_WDdi · 2023-10-29

**Soundness:** 3 good
**Presentation:** 3 good
**Contribution:** 2 fair
**Rating:** 6
**Confidence:** 2

**Summary:**

This paper introduces a framework named SURF, focusing on stochastic algorithm unrolling in federated learning contexts. The authors specifically employ descending constraints on the outputs of unrolled layers to make sure convergence. They also leverage the Lagrangian dual problem for optimization, with empirical validation on Graph Neural Networks (GNN).

**Strengths:**

1. The SURF framework stands out for its innovative method of implementing stochastic algorithm unrolling in federated learning. This novel approach, particularly the use of duality and gradient descent ascent in solving the Lagrangian dual problem, is a significant departure from traditional federated learning methodologies.
2. The paper provides a mathematical analysis of the convergence bound of SURF, indicating thorough theoretical underpinning. Also, the key technique of imposing descending constraints on the outputs of the unrolled layers to ensure convergence appears novel to me.

**Weaknesses:**

1. **Strong Assumptions**: The assumption of convexity in Assumption 1 is a significant limitation, given that many real-world scenarios involve non-convex functions. This assumption could restrict the applicability of the SURF framework in broader federated learning contexts.

2. **Lack of Comparative Analysis**: The paper does not provide an upper bound for the number of communication rounds needed to converge to a certain precision $\varepsilon$. This omission makes it difficult to compare SURF with other federated learning works, raising questions about the significance and practicality of the contribution.

**Questions:**

1. In the (SURF), there is no explicit representation of $\mathbf{W}_L = \boldsymbol{\Phi}(\boldsymbol{\vartheta}; \boldsymbol{\theta})$. Is this an intentional choice?
2. What is the complete formulation of the function $f$ in Assumption 2? Since the parameter of $f$ seems to depends not only on $\theta$ but also on other factors like $l, w_0$ etc., a clear definition is necessary.
3. Given that the fomula in (5) is based on expectations without explicit randomness, why does Theorem 2 require that (5) holds with a certain probability?

---

> ### Author Response · Authors · 2023-11-20
>
> We would like to thank the reviewer for their insightful review and suggestions, and for their positive evaluation of our work.
>
> 1. We thank the reviewer for their deliberate reading of our paper and theoretical analysis. The reviewer is indeed correct about the fact that the convexity assumption does not hold in practice. We acknowledge that we made an unintentional mistake in our write-up. Assumptions 1-5 are inherited from the constrained learning theory (Theorem 1 in [1]). This theorem requires strong duality of a version of (SURF) that learns a continuous function $\Phi$ instead of a parameterized one. The strong duality holds if $f$ is convex. Another sufficient condition for the strong duality to hold is by satisfying Assumptions 3 and 5, as proved in Proposition 3.2 in [1]. In our case, $f$ is definitely non-convex and, therefore, we had Assumptions 3 and 5 in place. These two assumptions are mild and hold for modern deep learning models. We have corrected the mistake in Assumption 1 in our revised manuscript and we thank the reviewer again for their deliberate reading of our paper.
> 3. We also would like to thank the reviewer for this insightful suggestion. We addressed this point in Theorem 3. The new theorem finds an upper bound of the gradient norm after a finite number of layers $L$. This allows us to find an upper bound on the number of layers required to achieve certain precision. This upper bound is derived in Appendix A.3.
>
> Questions:
> 1. The constraints in (Optimizer) are *implicit* constraints that are forced in the design of the unrolled netowrk $\Phi$. We wrote them explicitly in (Optimizer) only to help the comprehension of the idea behind unrolling. Observe that the two constraints were omitted from the Lagrangian function in (1).
> 2. We clarified Assumption 2 in our revised manuscript.
> 3. Theorem 2 does not require (5) to hold with probability. It is Theorem 1 that only guarantees that (5) holds with probability. Therefore, we had to incorporate this induced randomness in Theorem 2.
>
> [1] L. Chamon et al., Constrained learning with non-convex losses, IEEE Transactions on Information Theory, 2022.

---

### Official Review · Reviewer_94iK · 2023-10-30

**Soundness:** 4 excellent
**Presentation:** 4 excellent
**Contribution:** 3 good
**Rating:** 5
**Confidence:** 4

**Summary:**

This paper introduces a novel approach to accelerate FL convergence in a server-less setting. This is achieved via incorporating descending constraints on unrolled architectures. The proposed approach SURF is theoretically (Theorem 2) and empirically (Figure 2) substantiated. These findings demonstrate that an unrolled optimizer trained with SURF converges to a region close to optimality, ensuring its ability to generalize effectively to datasets within the distribution.

**Strengths:**

The paper is well written and the problem is well motivated. I find the descent constraints to arrive at a convergence guarantee very clever. As far as I'm aware, this method is novel, although I am not quite familiar with the L2O/unrolled algorithm literature so I can't say for sure.
Experiments are quite basic, but show some promising results.

**Weaknesses:**

There have been some existing works on serverless FL. For example, I find the following paper "FedLess: Secure and Scalable Federated Learning Using Serverless Computing" (Grafberger et al., 2021). I would suggest the authors to compare to some of these methods rather than standard FL approaches.

I also do not understand how the SURF method is limited to serverless FL. Can it be applied to standard FL instead?

It feels quite strange seeing that the accuracy curves of all other methods are very similar. Could it be due to this setting?

 I think Fig. 2 does not say anything about your convergence. What do accuracy and loss value at one point have to do with convergence guarantee? If anything, it would be Fig. 1, but it feels quite amazing to achieve perfect accuracy on CIFAR10 with only 20 training epochs. Can you please elaborate on what is happening in one communication round here?

Some of the experiment setup descriptions are quite vague. Could you elaborate on the following points:
- How were the other FL baselines modified to account for the serverless setting?
- What does it mean by "randomly-chosen agents are asynchronous with the rest of the agents". What is being asynchronous here, and how do you simulate it?
- What exactly is happening in one communication round?

**Questions:**

I have put my concerns in question form above.

---

> ### Author Response · Authors · 2023-11-20
>
> We would like to thank the reviewer for their insightful review and suggestions.
>
> We start by addressing the reviewer's comments:
> 1. We thank the reviewer for bringing up this insightful point. Our method, SURF, is not limited to serverless federated learning (FL) as the reviewer suggested. Standard FL is a special case of our setting when we consider a star graph, where a central node (i.e., server) is connected to all the other nodes and no communication is allowed between non-central nodes. Inspired by the reviewer's comment, we conducted extra experiments over star graphs and added them to the Appendix. For the sake of completeness, we also attached to the Appendix additional experiments over random graphs. We observe that regardless of the graph topology, we still achieve fast convergence compared to DGD and the FL benchmarks. However, our current implementation of the star-graph experiment only considers a homogeneous case, where the central agent is similar to the other agents, i.e., it learns a model based on its local data beside to its role as a server. To implement a classic FL with SURF, our algorithm should be modified in two ways: i) a different model update should be implemented in the server node, with the server having no data and just aggregating agents’ models, i.e.,
> $
> {\bf w}_{i,l} = [{\bf H}\_{l}({\bf W}\_{l-1})]_i
> $
> (Note the difference between this update rule and the one in (U-DGD)),
> and ii) a different test pipeline should be used, where the final server model is evaluated on a central test data.
> Additional experiments on this full classic FL setup are currently underway and we will add it to the camera-ready version of the manuscript. We expect the results to be similar to those in the star topology case, reported in Appendix B, since the difference between the two cases is subtle. We again thank the reviewer for his comment that inspired us to generalize our work to classic FL and we will update Section 5 in the camera-ready version with a new subsection on the extension to classic FL showing the update rule mentioned above.
> 2. Figure 1 does not imply that the other methods are quite similar. Zooming in on the benchmark curves would show a variability of +/-3% in the performance of these methods across the communication rounds. This variability is significant in real-world applications. Therefore, the figure only implies that the variability in the performance of the FL benchmarks diminishes when compared to **our method** that **is radically different**.
> 3. Figure 2 shows the effect of discarding the descending constraints. When the constraints are omitted as in standard unrolling, neither the objective/loss function $f$ nor the accuracy decreases monotonically over the layers. As shown in the figure, the final estimate has even jumped from 0% accuracy to 96% in one step raising doubts about the role of the previous layers. The figure shows that by adding the descending constraints, the estimates are **guaranteed** to descend over the layers. Figure 1, on the other hand, shows the convergence speed compared to other methods.
> 4. The remarkable fast convergence is attributed to the core idea of algorithm unrolling, which is to use data to learn **accelerated** descending directions. To describe the operations in one communication round, we emphasize that each agent aggregates its $K$-hop neighbors' models into a new model and then updates it with a local mini-batch. This process resembles one layer in our unrolled model. However, aggregating the models from each hop neighbor requires one communication round and therefore one layer requires $K$ communication rounds. In summary, each agent sends and receives data from $K$-hop neighbors in $K$ communication rounds and updates their own model once at every $K$ communication rounds. It is worth noting that this update rate is lower than the ones used in FL benchmarks, which update the model at every round. The fast convergence we experience in SURF is, therefore, due to the fact that we learn from training data how to weigh the neighbors' models efficiently and how to find a sequence of descending steps that converges quickly. This is not unique to our method, SURF, as similar fast convergence results have been reported for many other applications (see our Related Work Section and the references therein).

---

> > ### Author Response · Authors · 2023-11-20
> > **Part II**
> >
> > Finally, we address the questions raised by the reviewer:
> > 1. The FL baselines were not modified to account for the serverless setting. Having a server is a core concept in these methods that cannot be omitted. It is worth noting, however, that the serverless problem is a much harder problem than the standard one. This is observable in the convergence rates of DGD (Figure 1 Middle), which is a serverless method, and the FL benchmarks (Figure 1 Right). This significant difference in favor of the FL benchmarks occurs despite the fact that DGD allows all the agents to update their model at each communication round (i.e., using data from 100 agents), while the FL benchmark updates the model based on data coming from 5 agents only. This is the case because the data coming from the 100 agents does not contribute to learn one model but 100 models separately while forcing all the models to match each other. This requires more iterations to converge and deems a more challenging problem.
> > Nevertheless, to ensure more fairness in our comparisons, we, as mentioned earlier, have added experiments over star graphs that account for introducing a server to the network and disallowing any inter-communication between the agents. Figure 4 in the Appendix shows similar behavior to the one in Figure 1. This suggests that the graph topology does not contribute to the fast convergence.
> > 2. The asynchronous agents fail to update their estimates simultaneously with the other agents and, therefore, do not send a new update to their neighbors on time. Their neighbors hence use outdated models that have been received in an earlier round, which introduces errors to the process. At each communication round, we pick different agents to be asynchronous.
> > 3. We have described the operations executed in one communication round in point #4 above.

---

> > ### Comment · Reviewer_94iK · 2023-11-21
> > **Discussion**
> >
> > Thanks for replying to my questions.
> >
> > - Since model synchronization is not frequent (once every K comm rounds), I suspect your framework might suffer from client drift in a heterogeneous data setting. What do you think about this?
> > - I strongly believe FL without the client drift issue is just distributed learning (and in your case decentralized learning). Testing on client data simulated by a Dirichlet distribution has become a rather standard practice, so it would be quite hard to accept any FL method that did not demonstrate robustness against client drift.
> > - Regarding "The FL baselines were not modified to account for the serverless setting" -- This is why I suggested comparing against existing works on serverless FL, but it seems like my concern was not addressed.
> >
> > Overall, I don't think I am ready to accept your paper due to a lack of demonstration and will most likely stay my score. I however do think that the idea is promising and hope to see a future revision of it.

---

> > > ### Author Response · Authors · 2023-11-21
> > > **Response to additional questions by Reviewer 94iK**
> > >
> > > Thank you very much for your valuable comments and your engagement in the discussion. We agree that experiments on client drift are interesting and important additions to our work. We would love to have sufficient time to conduct these simulations, and we will make sure to include them in the camera-ready version of the manuscript. That being said, we respectfully do not think that their absence in the current version of the manuscript justifies the rejection of our paper, given its novel theoretical contributions and significant empirical gains over baseline methods.
> > >
> > > Your specific comments are addressed below:
> > >
> > > - **Model synchronization and client drift:** In a heterogeneous data setting, where updating the models every few communication rounds might be problematic in terms of client drift, we believe that our unrolled algorithm should respond much more quickly to such client drift than baseline methods which do not leverage unrolling. This is because convergence is much faster, and we are, therefore, more likely to respond well to changes in the optimal solution.
> > > - **Client drift experiments:** As we said above, this is indeed a valuable comment, and we thank you for pointing it out. Still, we do not agree that a paper on federated learning is not worth being accepted without client drift experiments. To incorporate your comment, we will add experiments on client drift to the final version of the paper to show the robustness of the proposed method to data heterogeneity.
> > > - **Serverless FL baselines:** Thank you for you clarification. What we meant here is that since the behavior of stochastic unrolled federated learning is fundamentally different from existing algorithms, we believe adding more baselines might not be worthwhile. Convergence of our method is orders of magnitude faster because of the use of unrolling. The drawback is that we are limited to training relatively small models. This tradeoff is well captured in existing simulations. Having said that, we will accommodate your suggestion by adding decentralized FL baselines, such as [A] and the references therein, to the camera-ready version of the paper.
> > >
> > >
> > >   [A] T. Sun, D. Li and B. Wang, "Decentralized Federated Averaging," in IEEE Transactions on Pattern Analysis and Machine Intelligence, vol. 45, no. 4, pp. 4289-4301, 1 April 2023, doi: [10.1109/TPAMI.2022.3196503](https://ieeexplore.ieee.org/abstract/document/9850408).
> > >
> > > We hope our response addresses your concerns. Please let us know if you have any additional questions or comments.

---

> ### Author Response · Authors · 2023-11-22
> **We updated the manuscript with new experiments**
>
> To follow up on our response, we have conducted the experiments under a Dirichlet distribution and compared our results to decentralized federated learning benchmarks, as requested by the reviewer. The results were added to Appendix B.3. The figure shows that our method is more robust to heterogeneity/client drift. SURF still converges fast, and the deterioration in the performance with the Dirichlet concentration parameter $\alpha$ is much slower than all the other benchmarks. This is the case since U-DGD, during its meta-training, learns to converge faster while balancing between the aggregated models received from the agents' neighbors and the local updates. This shows another strength of our method.

---

### Author Response · Authors · 2023-11-20
**General Response**

We would like to thank all the reviewers for their time and insightful comments and suggestions.

In this final remark, we would like to briefly address the main concerns of the reviewers and highlight the changes we made to our manuscript to resolve these concerns.

1. We admit that we made an unintentional mistake during the writing of the paper by claiming convexity in Assumption 1. As we clarified in our responses to reviewers WDdi and DuLq, who spotted this mistake, Assumptions 3 and 5 rule out the need for convexity. That was proven true in Proposition 3.2 and Theorem 1 in [1]. We, therefore, emphasize that we deal with non-convex loss functions and the goal of our method is to learn their stationary points. We accomplish this by forcing the gradient to descend over the unrolled layers.
2. Following the suggestion of Reviewer WDdi, we added Theorem 3 which shows the improvement in the gradient norm after a finite number of layers. Unlike the asymptotic analysis in Theorem 2, Theorem 3 can help derive an upper bound to reach certain precision. We derive this upper bound at the end of Appendix A.3.
3. We have conducted more experiments over different graph topologies. The first topology is a star graph, which is akin to the standard FL scenario that employs a centralized server. This new experiment addresses Reviewer 94iK's comment about fair comparisons as it shows a comparison between our method implemented with a server and the standard FL benchmarks and still confirms our findings. We also run the same experiment over random graphs and reported the results in Appendix B. From the figures, we observe that the fast convergence was achieved in all the experiments regardless of the graph topology.
4. We clarified our notation in Assumption 2 and Theorem 2 to make our message clear and concise based on Reviewer DuLq's insightful comments.

All the changes are highlighted in red in the revised manuscript.

[1] L. Chamon et al., **Constrained learning with non-convex losses**, IEEE Transactions on Information Theory, 2022.

---

### Meta-Review · Area_Chair_TWX6 · 2023-12-17

**Metareview:**

Summary: this paper proposes a variant to algorithm unrolling for a distributed federated learning setting. The main algorithmic novelty is a descent constraint, which they justify via an existing result.

Strengths: Algorithmic novelty

Weaknesses: the paper is dense and hard to read; it does too many things at the same time (federated learning, unrolling, stochastic GD, GNNs etc.). It would be very well served by distilling down the one key innovation in the cleanest setting, instead of losing clarity by trying to do too much. For example, just presenting it in the standard "star topology" FL setting and clarifying its differnces with pre-existing unrolling approaches. A brief background on algorithm unrolling would also useful, as this is not a very common notion.

The paper would be substantially improved by a clear and concise writing by providing the proper background on algorithm unrolling, and  first presenting the key algorithmic innovation in the simplest setting - ideally where it can be easily compared (visually and algorithmically) with other related approaches. The algorithm is novel and interesting but the paper is somewhat impenetrable.

**Justification For Why Not Higher Score:**

The paper is very densely written, it would really benefit from a brass-stacks rewrite.

**Justification For Why Not Lower Score:**

I have given it the lowest score, but wouldn't mind if it was bumped up.

---

### Decision · Program_Chairs · 2024-01-16

Reject